# CylinDeRS: A Benchmark Visual Dataset for Robust Gas Cylinder Detection and Attribute Classification in Real-World Scenes

**DOI:** 10.3390/s25041016

**Published:** 2025-02-08

**Authors:** Klearchos Stavrothanasopoulos, Konstantinos Gkountakos, Konstantinos Ioannidis, Theodora Tsikrika, Stefanos Vrochidis, Ioannis Kompatsiaris

**Affiliations:** Centre for Research & Technology Hellas, Information Technologies Institute, 57001 Thessaloniki, Greece; klearchos_stav@iti.gr (K.S.); kioannid@iti.gr (K.I.); theodora.tsikrika@iti.gr (T.T.); stefanos@iti.gr (S.V.); ikom@iti.gr (I.K.)

**Keywords:** cylinder, dataset, object detection, image classification, attribute extraction, deep learning

## Abstract

Gas cylinder detection and the identification of their characteristics hold considerable potential for enhancing safety and operational efficiency in several applications, including industrial and warehouse operations. These tasks gain significance with the growth of online trade, emerging as critical instruments to combat environmental crimes associated with hazardous substances’ illegal commerce. However, the lack of relevant datasets hinders the effective utilization of deep learning techniques within this domain. In this study, we introduce CylinDeRS, a domain-specific dataset for gas cylinder detection and the classification of their attributes in real-world scenes. CylinDeRS contains 7060 RGB images, depicting various challenging environments and featuring over 25,250 annotated instances. It addresses two tasks: (a) the detection of gas cylinders as objects of interest, and (b) the attribute classification of the detected gas cylinder objects for material, size, and orientation. Extensive experiments using state-of-the-art (SotA) models are reported to validate the dataset’s significance and application prospects, providing baselines for further performance evaluation and in-depth analysis. The results show a maximum mAP of 91% for the gas cylinder detection task and a maximum accuracy of 71.6% for the attribute classification task, highlighting the challenges posed by real-world scenarios and underlining the proposed dataset’s importance in advancing the field.

## 1. Introduction

In recent years, computer vision has witnessed remarkable advancements, enabling machines to perceive and understand visual information across a wide range of applications. Pivotal and well-established focus areas, including object detection and object attribute classification, have been extensively studied, while the availability of standardized datasets [1,2,3,4] has greatly facilitated the exploration of several research challenges associated with these tasks by serving as means to assess the comparative performance of different algorithms and techniques. Recently published deep learning techniques have shown that they can be highly effective and efficient in handling detection and classification tasks when coupled with image datasets with high-quality annotations [5,6,7]. This progress has led to significant breakthroughs in several fields, such as face detection and recognition, activity recognition, crowd analysis, as well as intelligent surveillance [8,9,10,11,12,13].

Apart from the need for datasets for typical visual tasks, the demand for domain-specific datasets is rising, driven by the increasing need for automation, continuous monitoring, and optimization of critical tasks across several domains, such as agriculture [14,15] and biomedical research [16,17]. In addition, domain-specific datasets affect the performance of machine learning models by improving their capability to accurately detect patterns and objects, while also enhancing their ability to generalize in unobserved content. An interesting and intriguing domain-specific application is related to the detection and analysis of cylinders, particularly used for transferring gases under low or high pressure, namely gas cylinders [18]. This application holds significant potential use cases in various operations, such as monitoring industrial facilities handling gas cylinders, preventing accidents associated with these items, as well as improving warehouse safety and storage efficiency [19,20]. In addition, the illegal trade and use of Ozone-Depleting Substances (ODSs) and hydrofluorocarbons (HFCs) included in gas cylinders, and the growing concern regarding the online commerce of hazardous substances that pose significant threats to the environment [21,22,23], require innovative and efficient solutions that can identify and mitigate the publishing of such advertisements on various digital platforms.

While significant progress has been made in generic object detection and attribute classification research [24,25], the existing datasets primarily focus on everyday scenes containing common objects [26,27,28,29,30]. This focus overlooks the specificity and diversity required for robust gas cylinder detection since the available datasets fail to account for the variations in gas cylinders’ material, size, and orientation, required for the identification of these attributes. In the context of domain-specific computer vision applications, synthetic data are widely used to compensate for the lack of real-world data [31]. However, there usually exists a notable gap between synthetic and real-world datasets in terms of quality, data bias, fairness, as well as potential ethical considerations and legal implications [32].

To address these limitations, we introduce CylinDeRS [33], a comprehensive domain-specific dataset for **cylin**der **de**tection in **r**eal-world **s**cenes and the classification of their attributes. The dataset contains a diverse compilation of images captured in various settings, ranging from industrial facilities and warehouse complexes to commercial spaces, covering both indoor and outdoor environments. The images feature gas cylinder objects in challenging conditions, including varying lighting scenarios, occlusions, and cluttered backgrounds, closely representing real-world situations. Figure 1 illustrates a selection of these images, highlighting the variety and complexity present in the dataset. The CylinDeRS dataset is publicly available from the Roboflow repository (https://universe.roboflow.com/klearchos-stavrothanasopoulos-konstantinos-gkountakos-6jwgj/cylinders-iaq6n (accessed on 17 January 2025)), enabling the research community to evaluate and extend its applications further.

The CylinDeRS dataset contains 7060 images and 25,260 annotated instances, making it a dataset of practical scale tailored to object detection and classification within the specialized domain of gas cylinders. Although not as extensive as some general-purpose object detection datasets, its domain-specific focus ensures high relevance and utility for this application. CylinDeRS distinguishes itself through its diversity, encompassing cylinders of varying sizes (short and long), materials (metal and fiber), and orientations (standing and fallen), often coexisting within a single image. This combination of material attributes, alongside semantic annotations for size and orientation, provide a more comprehensive representation than many SotA visual datasets. The dataset also introduces significant complexity, driven by the subtle distinctions among the included gas cylinders, thus making it challenging to achieve accurate detection and classification. Moreover, the presence in a single image of anywhere from one to one-hundred-forty-one cylinders with varying attributes further amplifies its difficulty, setting it apart from the existing object detection datasets. Captured in diverse real-world scenarios, the images feature cluttered environments, occlusions, and other challenges, underscoring the dataset’s value for advancing object detection and attribute classification tasks.

Every image in the CylinDeRS dataset is manually inspected and thoroughly annotated, with precise bounding box annotations outlining the contours of the gas cylinders using the Roboflow Annotate tool (https://docs.roboflow.com/annotate/use-roboflow-annotate (accessed on 17 January 2025)). Each gas cylinder instance within an image is further associated with three labels, indicating its attributes from a total of nine distinct classes, grouped as follows: material (metal, fiber, or unknown material), size (short, long, or unknown size), and orientation (standing, fallen, or unknown orientation). The two sets of annotations (bounding boxes and attribute classes) serve as ground truth labels for training and evaluating the performance of object detection and attribute classification models, respectively.

The entire dataset creation process, from the collection of images to the data pre-processing steps and the data annotation steps, is provided in the form of a well-defined methodology, which could serve as a guide for researchers aiming to create domain-specific datasets. To establish the application value of CylinDeRS, an extensive benchmark evaluation has been conducted using state-of-the-art (SotA) deep learning models trained on the proposed dataset. The experiments not only demonstrate the effectiveness of the dataset in terms of training deep learning models to automate practical gas-cylinder-related applications but also highlight the challenges that emerge with these tasks. Our contributions are summarized as follows:The introduction of CylinDeRS, a comprehensive dataset of 7060 images, with a total of 25,269 gas cylinder object instances, explicitly designed for gas cylinder detection and the classification of their key attributes (material, size, and orientation) in real-world visual scenes;The proposal of a systematic methodology for creating domain-specific datasets, including all the steps from data collection and pre-processing to data annotation;Extensive experiments using SotA object detection and attribute classification models, resulting in setting baselines and providing pre-trained benchmarks for further research, performance evaluation, and in-depth analysis.

The rest of the paper is organized as follows. Section 2 covers the related well-known publicly available datasets from the fields of object detection and object attribute classification. Section 3 presents the detailed methodology followed for the development of the dataset and defines its structure and characteristics. Section 4 describes the experimental setup, while Section 5 presents and discusses the evaluation results and the models’ performance. Finally, Section 6 concludes this work and outlines future steps.

## 2. Related Work

Datasets have always been pivotal resources for computer vision research. Within the domains of object detection and object attribute classification, the training of Deep Neural Networks (DNNs) is inseparable from various image datasets since they play a crucial role in developing DNN-based models for such tasks. Common object detection datasets, including (but not limited to) Pascal Visual Object Classes (Pascal VOC) 2012 [1], Common Objects in Context (COCO) [2], and Open Images [39], provide annotations related to the spatial position of objects within the image for a wide range of categories. Moreover, since visual attributes represent a significant portion of the details present within a scene, objects can be described by a diverse range of attributes that capture their visual appearance (color, texture, material) and geometry (size, shape, and orientation) [29]. To this end, widely used object attribute classification datasets, including COCO Attributes [3], Web Image Dataset for Event Recognition (WIDER) Attribute [40], iMaterialist [4], Visual Attributes in the Wild (VAW) [29], and Parts and Attributes of Common Objects (PACO) [30], incorporate comprehensive annotations, enriching the understanding of object attributes in various contexts.

### 2.1. Datasets for Visual Object Detection

Pascal VOC 2012 is a benchmark dataset featuring 11,530 images across 20 categories, with 27,540 Regions of Interest (RoIs) and 6929 segmentation annotations. Its straightforward categories and annotations have made it a fundamental resource for object detection and segmentation tasks. COCO comprises over 300,000 images annotated with 80 object categories and more than two million instances. Its diversity in scale, pose, and lighting conditions have established it as a valuable dataset for complex tasks such as object detection and panoptic segmentation. Open Images is one of the largest object detection datasets, containing 1.9 million images with 16 million bounding boxes for 600 categories. Its vast scale and extensive task support make it a versatile dataset for object recognition research.

### 2.2. Datasets for Visual Object Attribute Classification

COCO Attributes extends the COCO dataset by annotating 196 attributes for 180,000 objects across 84,044 images, resulting in 3.5 million object–attribute pairs. These annotations offer a nuanced understanding of objects’ visual and contextual properties. WIDER Attributes [40] focuses on human-specific characteristics, with 800,000 attribute labels across 13,789 images. Each bounding box includes 14 human attributes, such as age, clothing, and activity. The iMaterialist Fashion Attribute [4] dataset specializes in fashion-related tasks, offering over 1 million images annotated with 228 fashion attributes grouped into eight categories. It provides high-quality, fine-grained labels for tasks like attribute recognition and clothing recommendation systems. VAW [29] contains 72,000 images with 620 positive and negative visual attributes, resulting in 927,000 attribute labels, including color, shape, and texture, making it a valuable resource for attribute prediction.

While common object detection and object attribute classification datasets have played a vital role in advancing the field, they lack the specific characteristics required for the robust detection of gas cylinders and extraction of their attributes. To the best of our knowledge, there is a scarcity of dedicated datasets specifically designed for this purpose. The proposed CylinDeRS dataset fills this gap and aims to overcome the limitations of the existing publicly available datasets by providing a diverse collection of images capturing several arrangements of gas cylinders in challenging real-world scenarios.

## 3. The CylinDeRS Dataset Creation

In this section, we describe the steps followed for the creation of the CylinDeRS dataset. Initially, the process of exploring candidate image sources for data acquisition and selecting retrieval keywords is presented. Next, the data pre-processing steps are defined, and the annotation pipeline is thoroughly showcased. The ethical considerations regarding individual privacy and the employed anonymization process are addressed. Finally, the dataset’s structure is reported. These clearly defined steps also serve as a proposed methodology that could be used by researchers to construct their own domain-specific datasets.

### 3.1. Dataset Creation Methodology

Figure 2 illustrates the proposed methodology for creating domain-specific datasets. In the first step, candidate sources for the data acquisition process are identified. Each source undergoes exploration to identify relevant data (images and potentially available ground truth labels). In the next step, duplicate image data are removed using an image hashing algorithm, followed by manual review and removal processes, generating an updated image collection. The annotation process follows, involving meticulous curation through manual review, refinement of existing annotations, and addition of new annotations. Finally, the data are anonymized (by blurring identifiable faces) to address any privacy risks, resulting in the final list of images and ground truth annotations. This comprehensive process ensures the consistency and reliability of both image and annotation data, consequently promoting reproducibility and facilitating in-depth explorations and analyses of the dataset. Next, each of the main steps of this process are described in detail.

#### 3.1.1. Data Acquisition

The first step in the creation of the CylinDeRs dataset involves the collection of candidate images for gas cylinder detection and attribute classification. To this end, we identified potential sources by conducting extensive search, primarily focusing on publicly available repositories, including Kaggle (https://www.kaggle.com/ (accessed on 17 January 2025)), Roboflow (https://roboflow.com/ (accessed on 17 January 2025)), V7 Labs (https://www.v7labs.com/open-datasets (accessed on 17 January 2025)), and Google Datasets (https://datasetsearch.research.google.com/) (accessed on 17 January 2025). Through this exploration, Roboflow [41] was identified as the most suitable source for our specific scope as it offers publicly available datasets closely related to gas cylinder objects.

The proposed process of acquiring candidate images suggests utilizing a keyword-based search through synonyms, synonym sets, and English words linked to the prospective classes for the object detection and attribute classification tasks, resulting in more than fifty datasets [34,35,36,37,38,42,43,44,45,46,47,48,49,50,51,52,53,54,55,56,57,58,59,60,61,62,63,64,65,66,67,68,69,70,71,72,73,74,75,76,77,78,79,80,81,82,83,84,85,86,87,88,89,90]. In particular, the specific keywords we used are *“cylinder”*, *“gas cylinder”*, *“gas tank”*, *“gas cylinder detection”*, *“gas cylinder classification”*, *“short cylinder”*, *“long cylinder”*, *“oxygen cylinder”*, *“lpg cylinder”*, *“horizontal cylinder”*, *“fallen cylinder”*, *“steel cylinder”*, *“metal cylinder”*, *“fiber cylinder”*, *“metal tank”*, *“steel tank”*, and *“fiber tank”*. This approach yielded a diverse collection of 29,487 relevant images sourced from various settings, including industrial facilities, warehouses, and commercial spaces, covering a wide range of variations, accounting for diverse lighting conditions, occlusions, and cluttered backgrounds, reflecting the complexities encountered in real-world applications.

#### 3.1.2. Data Filtering

Following the download of all the candidate images and the respective pre-existing annotations (if any), the next step was an extensive filtering process, meticulously curating the images to maintain high reliability and quality. As part of this process, a Python (https://www.python.org/ (accessed on 17 January 2025)) script was employed to identify and remove duplicate images, streamlining the dataset for more efficient analysis. Initially, an MD5 hash-based approach [91] was utilized to eliminate redundant copies while storing one representative image from each set of duplicates. This step reduced the total number of images to 12,563. However, the manual inspection of the dataset images revealed that this approach occasionally retained multiple copies of nearly identical photos due to subtle variations, leading to distinct MD5 hash values.

To enhance the precision of duplicate identification, a more sophisticated approach was adopted, incorporating perceptual hashing, specifically the pHash (https://github.com/JohannesBuchner/imagehash (accessed on 17 January 2025)) algorithm. The updated script, which integrated the imagehash (https://pypi.org/project/ImageHash/ (accessed on 17 January 2025)) library, achieved more robust and accurate performance on discerning and managing similar images within the dataset, ensuring the preservation of distinct visual content while effectively eliminating redundancy. As a concluding step in the filtering process, the remaining images underwent manual review to identify and remove any duplicate images the algorithm may have overlooked. After this process, we concluded with a total collection of 7060 images.

#### 3.1.3. Data Annotation

The next step includes the annotation process of the curated collection of images through the inspection and refinement of the potentially pre-existing annotations and the addition of new annotations. A team of four researchers was employed, each responsible for manually annotating a subset of the dataset. The splits were overlapping, so it could be ensured that every image (and subsequently each instance) is annotated by at least two annotators. In case of a disagreement, a third annotator was involved to resolve it.

It is essential to highlight the distinction between the two tasks covered in this study: gas cylinder detection and gas cylinder attribute classification. For the detection task, the CylinDeRS dataset has been thoroughly annotated with precise bounding box annotations around the objects of interest, providing the location information (bounding box coordinates) and the corresponding class (i.e., “gas cylinder”) for every gas cylinder instance in the image. For the attribute classification, each instance is associated with three labels describing its material, size, and orientation attributes. Currently, CylinDeRS supports three attributes with a total of nine classes: material (metal, fiber, and unknown material), size (short, long, and unknown size), and orientation (standing, fallen, and unknown orientation).

The annotation process for each image and the respective instances followed a systematic procedure consisting of three essential steps:**Inspection:** Firstly, a comprehensive manual inspection of the pre-existing annotations should be considered, as provided by the original datasets gathered during the data acquisition step (Section 3.1.1). In our case, this critical examination aimed to evaluate the initial accuracy and reliability of the bounding box annotations. Moreover, the process involves a thorough examination of the dataset to identify images where gas cylinders lack bounding box annotations. Any duplicate annotations of the same instance are removed, and potential discrepancies or errors in the existing annotations are identified, establishing a solid foundation for subsequent refinement and the creation of new annotations. Overall, a total of 448 duplicate bounding boxes were identified, 672 bounding boxes were missing, and 3239 instances contained errors in the existing annotations since the bounding boxes were not perfectly aligned with the instances, requiring resizing or adjustments in position (left, right, up, or down).**Refinements:** The next stage revolves around refining the size and labels of the pre-existing bounding boxes. In CylinDeRS, a diligent process was followed where the bounding boxes were manually adjusted to precisely encapsulate the gas cylinder instances, maximizing their accuracy and adherence to the true boundaries of the objects. Furthermore, the original labels associated with the pre-existing bounding boxes were refined, ensuring they accurately represent the class of “gas cylinder”, which is in scope for the proposed dataset’s object detection task. This iterative refinement process elevated the precision and consistency of the annotations, further enhancing the dataset’s overall quality.**New annotations:** The final stage of the annotation process resides in providing new annotations for the objects of interest. This process involves generating new annotations, serving a two-fold purpose: rectifying missing bounding box annotations for gas cylinder objects and assigning instance-level labels for the attribute classification task. Initially, the process includes the annotation of gas cylinder instances in images lacking annotations as well as in those with partial annotations, as identified through the inspection step. The boundaries of the gas cylinders are precisely delineated, accompanied by the corresponding “gas cylinder” class. Each instance is associated with an additional set of three labels representing the gas cylinder’s material, size, and orientation attributes, considering the overall context of the image. The new annotations contribute to a comprehensive dataset that accurately depicts the gas cylinders’ spatial arrangement, also providing detailed information about their attributes.

#### 3.1.4. Ethical and Legal Considerations

The CylinDeRS dataset, while primarily focused on gas cylinder objects, inadvertently features instances of individuals. Through a rigorous manual inspection of the entire dataset, individuals were identified in 800 out of the 7060 images. Considering that all collected images sourced from the datasets [34,35,36,37,38,42,43,44,45,46,47,48,49,50,51,52,53,54,55,56,57,58,59,60,61,62,63,64,65,66,67,68,69,70,71,72,73,74,75,76,77,78,79,80,81,82,83,84,85,86,87,88,89,90] are licensed under the CC-BY 4.0 license, enabling derivative works, the ethical considerations were explored, particularly regarding individual privacy. In adherence to legal frameworks and ethical principles, we addressed potential identifiability concerns and privacy risks associated with the dataset. As part of our commitment to responsible data usage, all identifiable faces within the dataset exhibiting clear details have been anonymized using a blurring technique. This precautionary measure mitigates the risk of unintentional identification, ensuring compliance with data privacy regulations and ethical standards. It is important to note that anonymization was applied selectively; for faces with limited visibility, anonymization was deemed unnecessary to maintain the dataset’s utility for gas cylinder object analysis while still adhering to ethical standards. This balanced approach emphasizes our commitment to responsible data management.

#### 3.1.5. Challenges

The creation of the CylinDeRS dataset involved several challenges that impacted both the annotation process and the overall representativeness of the dataset. One major challenge was the annotation complexity, especially when dealing with images containing multiple gas cylinders or instances where the gas cylinders were occluded. Drawing accurate bounding boxes around the gas cylinders and assigning the correct attributes required extensive manual review. Another challenge was the variability in gas cylinder appearance as gas cylinders differ widely in shape, size, and material depending on their use and environment. Cylinders in industrial settings, for example, are often stored in groups, stacked, or partially obscured by other objects, necessitating careful selection of images and data augmentation strategies to ensure a representative sample. Environmental factors also played a role, with the surrounding context—such as lighting, background clutter, or the presence of large objects—affecting the apparent size and orientation of gas cylinders. For instance, a gas cylinder might appear smaller if placed next to a large building or vehicle, requiring semantic analysis to accurately determine its size. Lastly, the dataset faced a data imbalance issue as some attribute categories, such as “fallen” and “unknown” orientations, had fewer instances compared to more common categories like “standing” or “short”. This imbalance could introduce bias in model training, making it more challenging for algorithms to accurately detect and classify less frequent scenarios.

### 3.2. CylinDeRS Dataset

The result of the aforementioned steps is an extensive collection of 7060 images, each featuring a variable number of gas cylinder objects. In particular, CylinDeRS contains a total of 25,269 instances of gas cylinders, with an average of 3.6 instances per image. Table 1 provides an overall understanding of the dataset’s scale and presents a comprehensive overview of its structure. It details the distribution of images and gas cylinder instances as established after the annotation process, ensuring a fair distribution of both the number of images and instances among training, validation, and test sets. The table also includes the average number of gas cylinders per image for each set. Notably, following a 70:20:10 split ratio, the dataset comprises 4915 training images, 1434 validation images, and 711 test images, with 18,137 instances in the training, 4862 in the validation, and 2270 gas cylinders in the test set.

Figure 3 showcases a selection of representative samples for the two supported tasks, providing a glimpse into the dataset’s content and highlighting its varied and complex nature. In particular, Figure 3a illustrates indicative samples regarding the gas cylinder detection task, where each cylinder object is localized and the corresponding bounding box is delineated. Beyond bounding boxes, the dataset includes attribute annotations for each gas cylinder instance. Figure 3b showcases representative samples for the supported attributes used in the attribute classification task. These annotations delve into various aspects and characteristics, enhancing the dataset’s utility and expanding its applicability.

To comprehensively characterize gas-cylinder-related objects, we defined three key attributes based on [92], each with at least three associated classes:

**Material:** The CylinDeRS dataset covers three classes in terms of the material attribute, “metal”, “fiber”, and “material unk(nown)”, to represent distinct compositions of gas cylinders. Cylinders in the “metal” class are primarily composed of metallic materials, such as steel or aluminum [92] (p. 73), exhibiting robustness, strength, and durability, making them suitable for various industrial, commercial, and medical settings. The “fiber” material class comprises gas cylinders consisting of an inner container that is over-wrapped with durable lightweight fiber-based materials (glass, aramid, or carbon), offering a high strength-to-weight ratio [92] (p. 70). They find applications in scenarios where weight reduction is essential, such as aerospace or portable gas storage for outdoor activities. The “material unk(nown)” class encompasses gas cylinders that may lack distinct features and exhibit irregularities or burn marks, complicating the identification of their material composition.

**Size:** The size attribute corresponds to three classes, “short”, “long”, and “size unk(nown)”, representing variations in the physical dimensions of gas cylinders and providing granularity for size-based categorization. It is important to emphasize that defining the size attribute of an object within an image requires semantic analysis of the content as the surrounding scene significantly contributes to the contextual interpretation of the object’s size. For instance, a gas cylinder next to a skyscraper will appear smaller than the same cylinder next to a bicycle. The CylinDeRS dataset considers this semantic information during annotation, enabling accurate size attribute determination. Gas cylinders categorized as “short” have relatively compact vertical length (height) dimensions with a low Length-to-Diameter ratio (L/D), primarily used for low-pressure liquefied gas services [92] (p. 70). They are typically more maneuverable and more accessible to handle in environments with spatial constraints, finding application in diverse settings such as domestic use and laboratories. Conversely, “long” instances in the CylinDeRS dataset exhibit taller dimensions compared to their “short” counterparts, with a high L/D ratio accommodating larger volumes of gases. They are generally used for high-pressure non-liquefied gas service [92] (p. 70) and are usually well suited for applications requiring extended usage before replacement or refill, such as industrial processes and manufacturing. Within the “size unk(nown)” class, the instances predominantly feature occluded gas cylinders, resulting in difficulty in determining their size. These occlusions are usually due to structural obstacles (walls, beams, pillars, etc.), stacking/overlapping arrangements (stored in storage facilities, transportation vehicles, etc.), or partial obstruction by other objects.

**Orientation:** Three classes are supported for the orientation attribute, “standing”, “fallen”, and “orientation unk(nown)”. Similarly to the size attribute, orientation is inherently a contextual feature, with its determination depending on the semantic interpretation of the given scene. In CylinDeRS, semantic information related to the orientation of the gas cylinders is incorporated during annotation, wherein the context of the entire image is taken into consideration before assigning labels to specific instances within the image. Gas cylinders labeled as “standing” are positioned upright, with their longitudinal axis aligned vertically. They are typically secured stably, ensuring proper containment of pressurized gases [92] (pp. 186, 228, 494, 525). Gas cylinders categorized as “fallen” are depicted in a horizontal or tilted orientation, suggesting that they are not upright and have deviated from their standard standing position. This class indicates instances of gas cylinders potentially posing safety concerns due to the risk of uncontrolled movement or gas release.

Table 2 provides detailed statistics regarding the different attribute categories. As depicted in the table’s last column, the “metal” category dominates the material attribute with a total of 20,514 instances, with “fiber” gas cylinders depicted in 4518 instances, while the “material unk” category exhibits comparatively lower counts of 237. Within the size attribute, the total instances labeled as “short” are 13,450, followed by “long” with 6528 and ”size unk” with 5291. For the orientation attribute, the total “standing” instances, 23,311, significantly outnumber the others, with the “fallen” and “orientation unk” classes depicted in 1379 and 579 instances, respectively.

The scarcity of instances in the “material unk”, “fallen”, and “orientation unk” categories within the CylinDeRS dataset can be attributed to the relatively uncommon nature of these scenarios in real-world settings. Burnt or damaged gas cylinders (“material unk”) are uncommon in typical industrial and warehouse settings. Similarly, safety regulations minimize instances of “fallen” gas cylinders that are generally securely positioned and adherent to safety standards to prevent accidents. Furthermore, the limited instances within the “orientation unk” class primarily involve gas cylinders where the visible cues and the semantic context necessary for accurate orientation determination are limited. This is an unusual scenario in real-world cases. The imbalances in these categories reflect the dataset’s fidelity to real-world occurrences, where certain situations are less frequent, making them crucial for training models to also recognize and handle such uncommon scenarios effectively.

## 4. Experimental Setup

This section delves into the details of the conducted experiments, covering the object detection and attribute classification algorithms selected for training and evaluation using the introduced gas-cylinder-related dataset. The primary objective of these experiments is to assess the effectiveness and robustness of the selected models in detecting gas cylinders and classifying their attributes, with a particular focus on enhancing their practical utility in real-world scenarios. Moreover, the implementation and training processes are presented, along with the augmentation techniques and the hardware and software configurations. Finally, the metrics adopted for evaluating the models are discussed.

### 4.1. Gas Cylinder Detection

For the gas cylinder detection experiments, SotA object detection algorithms are selected from three different object detector categories in order to explore a wide range of architectures: Convolutional Neural Network (CNN)-based two-stage methods, CNN-based single-stage approaches, and transformer-based architectures. The representative models Faster R-CNN [93], YOLOv8 [94], YOLOv11 [95], and RT-DETR [96] are utilized for each respective category:

**Faster R-CNN** [93] integrates a Region Proposal Network (RPN) into its architecture, improving object detection performance. For experiments, we use Faster R-CNN with ResNet-101 [97] and Feature Pyramid Network (FPN) [98], which enhances detection, especially for smaller objects.

**YOLOv8** [94] is a unified model that predicts bounding boxes and object classes in a single pass. It features components like the Backbone, Spatial Pyramid Pooling Fusion (SPPF) layer, and C2f module for high accuracy and speed. We use the “YOLOv8m” version for its favorable trade-off between performance and computational cost.

**YOLOv11** [95] is the latest model in the YOLO family, combining adaptive attention mechanisms and improved feature fusion for real-time performance and precise detection across various object scales. The “YOLOv11m” model is used in our experiments.

**RT-DETR** [96] leverages transformer-based multi-head attention for real-time object detection. It uses an efficient hybrid encoder for multi-scale feature interaction. We use the “rtdetr-l” implementation for consistency with YOLOv8 and YOLOv11 in terms of model comparison. Upon acceptance of this work, all models trained for the gas cylinder detection task on the proposed CylinDeRS dataset will be made accessible to the community.

### 4.2. Gas Cylinder Attribute Classification

Regarding object attribute classification, CNNs have also been the predominant selection in recent years [99,100]. Object attribute prediction is formally characterized as a multi-label classification task, requiring the prediction of all attributes associated with a given object [29,30,100]. However, the multi-label approach is unsuitable for the objective of this work, which requires the prediction of exactly three labels per gas cylinder instance—one for each attribute. This requirement arises from the constraint that an object cannot simultaneously exhibit more than one class of the same attribute. Moreover, the conventional multi-class classification method is not applicable due to its inherent design for assigning a single label per object that contradicts the core requirement of CylinDeRS, which necessitates the prediction of three labels (material, size, and orientation) for each instance.

To overcome these issues, we adopted a multi-head multi-class classification approach for this task that enables the extraction of three labels for each gas cylinder instance, resulting in a single prediction for each attribute type, in one pass (see Figure 4). By approaching this as a multi-class classification task with three different output heads, the conducted experiments involve the exploration and modification of SotA deep-learning-based classifiers. To provide a set of widely used pre-trained models to the research community and for implementation simplicity, the widely used ResNet family [97] of classifiers, specifically ResNet-50 and ResNet-101, are selected and adapted to effectively support this multifaceted task.

**ResNet-50** [97] is a pivotal member of the ResNet family. As a variant of the ResNet architecture with 50 layers, it has established itself as a benchmark in computer vision tasks. Employing skip connections and forming residual blocks, ResNet-50 facilitates the smooth flow of information through the network and excels in capturing representative features. **ResNet-101** [97] is another member of the ResNet family that extends the architecture to 101 layers. It builds upon the success of ResNet-50, utilizing a larger number of residual connections and blocks, providing a deeper model. The additional layers contribute to a more comprehensive understanding of image features, potentially improving accuracy, especially in scenarios with complex details or subtle variations.

In the experiments, the PyTorch implementation (https://pytorch.org/vision/main/models/resnet.html (accessed on 17 January 2025)) of the ResNet-50 and ResNet-101 models is deployed, where the stride for downsampling is placed to the second 3×3 convolution. Upon acceptance of this work, both models trained for the attribute classification task using the proposed CylinDeRS dataset will be made publicly available.

### 4.3. Implementation and Training

To achieve improved performance, transfer learning is employed for the selected models for both object detection and object attribute classification tasks. Additionally, during the training process, we carefully tune a range of hyperparameters for both tasks, including the learning rate, batch size, momentum, weight decay, and optimization algorithms, through an extensive validation process. This leads to reporting optimal performance with an ideal balance between convergence speed and generalization capability for the training models that are evaluated in this work.

In order to conduct a fair comparison, all models for the gas cylinder detection task are pre-trained on the widely used COCO dataset. In terms of hyperparameters, the learning rate is initially set to 0.01 and decayed by a factor of 0.1 at the 50th and 100th epochs. The total number of training epochs is 200. The models are trained using Stochastic Gradient Descent (SGD) [101] as the optimizer with a batch size of 32, momentum of 0.9, and weight decay of 0.001. The deep learning models for the gas cylinder detection task are trained using the training set and evaluated on the validation and test sets of CylinDeRS; the evaluation results are summarized below in Table 3 and discussed in Section 5.

For the attribute classification task, the selected models are fine-tuned using pre-trained ImageNet weights to establish a benchmark comparison. The training process runs for a total of 150 epochs, with an initial learning rate of 0.001, which decays by a factor of 0.1 every 40 epochs. The Adam optimizer [102] is used, with a batch size of 32. To prepare the dataset for attribute classification, all cylinder objects are extracted from the images using ground truth bounding boxes, creating a collection of 25,269 images resized to 224×224 pixels for model input. The models are trained on the CylinDeRS training set, with their performance evaluated on both the validation and test sets; the evaluation results are summarized below in Table 4 and discussed in Section 5.

It is important to note that padding is applied to preserve the original features of the gas cylinder instances, preventing the neural network from learning potentially inaccurate features during training. This strategic use of padding ensures the network’s understanding of the spatial characteristics and contextual relationships within and around each cylinder. During convolution operations, the input image undergoes a series of convolution filters, potentially shrinking its spatial dimensions (height and width) [103]. Padding addresses this issue by adding a border of zeros around the original image. This maintains the original spatial dimensions throughout the network, enabling the CNN to capture features from the entire image. Without padding, these features might be distorted due to the shrinking size, potentially leading to inaccurate feature learning and classification errors. Furthermore, the edge pixels might be excluded from the convolution operation, causing the network to miss these potentially informative details. This can lead to the network learning inaccurate features that are biased towards the center of the image, ultimately impacting classification performance.

### 4.4. Data Augmentation

To address the issue of data imbalance and enhance the generalization capability of the models, particularly for underrepresented classes (i.e., “fallen” and “material unknown”), data augmentation pre-processing techniques were employed. These techniques aim to artificially increase the diversity of the training dataset, reducing the risk of overfitting and improving the model’s robustness. These techniques include random cropping and horizontal flipping as the geometrical transformations, and contrast and brightness enhancement as the intensity adjustments. A random crop is an arbitrary sample of the original image. The randomly cropped portion is resized to the original image size and fed to the network. Horizontal flipping involves flipping the image horizontally 180 degrees to increase the diversity of the samples. The contrast enhancement method expands the range between the image’s lightest and darkest pixels, resulting in a more pronounced distinction between features and details. Brightness enhancement focuses on adjusting an image’s overall luminance or brightness level for better visual perception.

### 4.5. Hardware and Software Configurations

All experiments were conducted on a computer system with Ubuntu 20.04 equipped with an Intel Core i7 3.6 GHz CPU, 128 GB RAM, and an Nvidia RTX 3090 GPU with 24 GB VRAM. The software stack consists of the Python language, along with popular computer vision libraries such as OpenCV (https://opencv.org/ (accessed on 17 January 2025)), TensorFlow (https://tensorflow.org/ (accessed on 17 January 2025)), and PyTorch (https://pytorch.org/ (accessed on 17 January 2025)). The deep learning models are trained and evaluated with GPU acceleration, harnessing the enhanced processing capabilities for efficient training and inference.

The training times for the object detection models on the CylinDeRS dataset vary depending on the architecture. Faster R-CNN required approximately 12 h, while YOLOv8m demonstrated faster training, taking around 4 h. YOLOv11m showed a similarly efficient training time of approximately 3 h and 30 m, reflecting its optimized architecture for quicker convergence. RT-DETR, with its transformer-based architecture, took longer, requiring around 6 h and 30 m due to the additional computational overhead. For the attribute classification task, ResNet-50 required approximately 5 h for the training process, while ResNet-101 took around 7 h, reflecting the increased complexity of the deeper ResNet-101 architecture compared to ResNet-50.

### 4.6. Evaluation Metrics

To assess the performance of the gas cylinder detection algorithms, the evaluation metrics commonly used in object detection tasks are adopted: precision, recall, and mean average precision (mAP). Precision measures the accuracy of positive predictions made by an object detection model, while recall quantifies the ability of an object detection model to find all relevant objects in the provided data. The mAP metric evaluates the overall performance of an object detection model across multiple object classes. It combines both precision and recall by calculating the area under the precision–recall curve—known as average precision (AP)—for each class and then taking the mean of these AP scores. The values are calculated as depicted in the following equations:(1)$$\mathrm{Precision}=\frac{TP}{TP+FP},\mathrm{Recall}=\frac{TP}{TP+FN},\mathrm{mAP}=\frac{1}{\left|Classes\right|}\sum_{i=1}^{\left|Classes\right|}AP_{i}$$
where TP, FP, FN, and AP denote the true positive, false positive, false negative, and average precision values, respectively.

For the evaluation of the attribute classification models, the accuracy evaluation metric is utilized. Accuracy is a highly intuitive way to evaluate the performance of any classification algorithm by calculating the percentage of the correct predictions. Specifically, it corresponds to the division of the number of correct predictions with the total number of predictions, as shown below:(2)$$\mathrm{Accuracy}=\frac{TP+TN}{TP+FP+TN+FN}$$
where TP, TN, FP, and FN denote the true positive, true negative, false positive, and false negative values, respectively.

## 5. Results and Discussion

In this section, the evaluation of the performance of SotA models on the CylinDeRS dataset is presented regarding both gas cylinder detection and attribute classification tasks. By conducting this evaluation, we aim to understand the strengths and limitations of the models and identify potential areas for improvement, which is essential for advancing the accuracy and reliability of automated systems for real-world applications. The results for each analysis are thoroughly presented, and a comprehensive discussion unfolds around the outcomes.

### 5.1. Gas Cylinder Detection Results

We initiate the evaluation by conducting a series of experiments to compare the performance of SotA algorithms on the CylinDeRS dataset. Four object detection methods were selected, Faster R-CNN [93], YOLOv8 [94], YOLOv11 [95], and RT-DETR [96] (as introduced in Section 4.1), each representing a unique category in terms of how they handle and process images.

#### 5.1.1. Performance Evaluation

The summary of the quantitative evaluation results, employing the three selected state-of-the-art models and various metrics, is presented in Table 3. The evaluated deep-learning-based models report adequate results in terms of precision, recall, and mAP regarding both the validation (Val) and test (Test) sets. YOLOv11 exhibits the highest performance across all the metrics, achieving a notable precision of 0.915 on the validation set and 0.905 on the test set, alongside strong recall values of 0.853 (Val) and 0.816 (Test), resulting in high mAP scores of 0.923 (Val) and 0.910 (Test). This suggests that YOLOv11 maintains consistency across both dataset splits, reinforcing its reliability. Meanwhile, YOLOv8 and RT-DETR also exhibit strong performance, with slightly lower precision than YOLOv11 but maintaining competitive mAP scores of 0.916 and 0.922 (Val) and 0.904 and 0.903 (Test), respectively, highlighting their balanced precision–recall profiles. In comparison, Faster R-CNN shows relatively moderate performance. With a validation precision of 0.640 and a test precision of 0.610, alongside recall values of 0.568 (Val) and 0.550 (Test), this two-stage model may benefit from enhancements, particularly in terms of recall, on both sets. Overall, the quantitative analysis reveals that, within the proposed dataset, the single-stage approach for object detection outperforms the two-stage approach and exhibits a performance advantage over the transformer-based method.

To gain deeper insights into the performance of the fine-tuned models, qualitative results are also presented in Figure 5. Example images from the CylinDeRS dataset are showcased along with the corresponding detection results for each of the models. In the left column, the ground truth bounding box annotations are highlighted in yellow color, delineating the boundaries of each gas cylinder instance in the showcased images from the CylinDeRS dataset within the image. The remaining three columns illustrate the predicted output for each of the explored models, featuring bounding boxes around detected gas cylinder instances, accompanied by the corresponding mAP scores. We can observe that Faster R-CNN exhibits moderate performance with lower confidence scores and a higher number of false positives. YOLOv8 and YOLOv11 demonstrate the best overall performance, accurately detecting most gas cylinder instances with high confidence scores. RT-DETR also performs well, achieving results similar to the YOLO models, with high confidence scores, albeit missing detecting a challenging instance.

**Table 3 sensors-25-01016-t003:** Gas cylinder detection performance comparison of Faster R-CNN, YOLOv8, YOLOv11, and RT-DETR fine-tuned using the CylinDeRS dataset.

Model	Precision			Recall			mAP	
	Val	Test		Val	Test		Val	Test
Faster R-CNN [93]	0.640	0.610		0.568	0.550		0.612	0.590
YOLOv8 [94]	0.915	0.902		0.850	0.812		0.916	0.904
YOLOv11 [95]	0.915	0.905		0.853	0.816		0.923	0.910
RT-DETR [96]	0.913	0.884		0.843	0.825		0.922	0.903

#### 5.1.2. Limitations

To identify the limitations and challenges within the target domain, a thorough qualitative examination is conducted, revealing the following key findings: (a) detection accuracy may be compromised in instances of severe occlusion or partial concealment of gas cylinders behind objects, (b) the models might encounter difficulties in handling unusual settings, including highly reflective surfaces or non-standard gas cylinder shapes, and (c) attention is required for instances of false positive detections, where objects are incorrectly identified as gas cylinders mostly due to objects with similar shapes or appearances, such as bottles, pipes, or cylindrical containers. All these cases can have significant implications in critical applications, where misidentification might lead to potential risks or errors.

Figure 6 provides a visual representation of instances where these limitations become apparent. The top row illustrates representative—to each limitation—samples highlighted by the bounding box annotations, as included in ground truth annotation files of the CylinDeRS dataset, while, in the second row, image samples of the detections extracted using the YOLOv11 model are depicted. In the last row, the reason for the erroneous prediction is depicted. The first sample reveals instances where occlusion led to failure in identifying certain gas cylinders (scenario (a)). In the second sample, a false positive detection is performed due to a gas cylinder instance reflection on a highly reflective surface (scenario (b)). The third and fourth samples depict false positive detections of objects with similar shapes as gas cylinders (scenario (c)).

### 5.2. Gas Cylinder Attribute Classification Results

The evaluation process is applied on object attribute classification, employing the CylinDeRS dataset to rigorously assess the accuracy of prominent deep learning models on the material, size, and orientation attributes. In this domain, we focus on two distinguished members of the ResNet family—ResNet-50 and ResNet-101—both known for their commendable performance in feature extraction and representation learning.

#### 5.2.1. Performance Evaluation

The results presented in Table 4 include the comparative performance of ResNet-50 and ResNet-101 for the performance evaluation of the attribute classification task on both the validation and test sets. Notably, the individual accuracy scores for material, size, and orientation provide insights into the models’ effectiveness in discerning each specific attribute. ResNet-50 exhibits commendable accuracy scores of 0.946 (Val) and 0.956 (Test) in material and 0.932 (Val) and 0.954 (Test) in orientation classification but demonstrates relatively lower Val and Test accuracy of 0.735 and 0.753 in the size attribute predictions. On the other hand, ResNet-101 showcases improved performance across all the attribute categories, achieving higher accuracy values in the material classification, with scores of 0.959 (Val) and 0.962 (Test), and shows similar improvements in the size attribute, scoring 0.751 (Val) and 0.772 (Test). Additionally, its orientation accuracy is slightly higher than ResNet-50’s, reaching 0.944 (Val) and 0.958 (Test).

Since the individual attribute scores reflect the model’s predictive accuracy for each attribute separately, the overall accuracy metric is incorporated for a more accurate evaluation, providing the model’s performance across all three attributes. In this case, a true positive result indicates that the material, size, and orientation attributes were correctly predicted for a gas cylinder. The overall accuracy metric reflects each model’s holistic effectiveness in accurately identifying all three attributes concurrently. In this regard, ResNet-101 leads, with an overall accuracy of 0.704 on the validation set and 0.716 on the test set, representing a performance boost compared to ResNet-50, which scores 0.670 (Val) and 0.675 (Test).

To further assess the performance of the attribute classification models fine-tuned on the proposed dataset, qualitative results are showcased in Figure 7. The models are provided with input in the form of images derived by cropping the gas cylinder instances from the original images based on the ground truth bounding boxes. The visualization includes sample gas cylinder instances and their corresponding material, size, and orientation classifications predicted by the models. The top row presents the input images alongside their ground truth attribute labels. The predictions for ResNet-50 and ResNet-101 are displayed in the middle and bottom rows.

**Table 4 sensors-25-01016-t004:** SotA performance comparison across the different attribute categories of the CylinDeRS dataset.

Model	Material			Size			Orientation			Overall	
	Val	Test		Val	Test		Val	Test		Val	Test
ResNet-50 [97]	0.946	0.956		0.735	0.753		0.932	0.954		0.670	0.675
ResNet-101 [97]	0.959	0.962		0.751	0.772		0.944	0.958		0.704	0.716

#### 5.2.2. Limitations

The exploration of limitations is crucial for a comprehensive understanding of the attribute classification outcomes within the CylinDeRS dataset. After a thorough qualitative evaluation, two main limitations have been observed: (a) one notable limitation stems from the inherent complexity of gas cylinder attributes and the data imbalance due to unusual scenarios (e.g., burnt or heavily damaged gas cylinder), as described in Section 3.2, and (b) the occurrence of ambiguous instances, such as gas cylinders with uncertain material compositions or obscured orientations, posing challenges for accurate classification.

Figure 8 illustrates misclassified visual samples. The first sample is a case of size attribute misclassification, where both models inaccurately predict the size attribute as “short” instead of “size unk”, possibly attributed to misleading visual cues resulting from the heavily cropped gas cylinder instance. Similarly, in the second example, ResNet-50 categorizes the gas cylinder object as “short” instead of “size unk”. This could be due to the resemblance of this instance to a short round-shaped cylinder. In the last image, the ResNet-50-based method struggles to classify size and orientation attributes, while ResNet-101 predicts all the attributes correctly. With fewer layers, ResNet-50 might primarily focus on learning lower-level features like edges and corners, which might be insufficient for tasks requiring understanding the object’s size and spatial position within the scene.

For further interpretation of the results and analysis of the limitations for the attribute classification task, the confusion matrices for the material, size, and orientation attributes are presented in Figure 9. These matrices illustrate a detailed and comprehensive examination of the top-performing model’s results for each class compared to the actual labels.

Regarding the **material attribute**, for instances genuinely belonging to the “metal” class, the model predicts three-thousand-eight-hundred-one (96.3%) instances correctly; however, there are one-hundred-thirty-nine (3.5%) instances misclassified as “fiber” and eight (0.2%) misclassified as “material unk”. For the “fiber” class, the model predicts seven-hundred-eighty-five (91%) correctly, but there are seventy-three (7.5%) instances misclassified as “metal” and four (0.5%) as “material unk”. For instances of the “material unk” class, the model correctly predicts only four (7.7%) cases labeled as “material unk”. There are thirty-three (63.5%) instances misclassified as “metal” and fifteen (28.8%) misclassified as “fiber”. The high misclassification rates for the “material unk” class reflect that the model’s ability to discern this category is limited, which suggests a potential need for additional data to increase its accuracy in identifying such instances.

For the **size attribute**, the model correctly identifies 1846 (72.3%) instances labeled as “short”. There are 468 (18.3%) instances misclassified as “long” and 239 (9.4%) misclassified as “size unk”. Similarly, the model correctly predicts 998 (77.8%) instances labeled as “long”. There are 191 (14.9%) instances misclassified as “short” and 94 (7.3%) as “size unk”. The model correctly identifies 681 (66.4%) instances labeled as “unknown size.” There are 146 (14.2%) instances misclassified as “short” and 199 (19.4%) misclassified as “long”, revealing challenges in correctly assigning instances to the “size unk” class. The results indicate that, while the model performs relatively well in distinguishing between “short” and “long” instances, it struggles with the “size unk” category, potentially due to insufficient or ambiguous features distinguishing it from the other two classes.

In regard to the **orientation attribute**, the model performs well in correctly identifying instances labeled as “standing”, with a high count of 4275 (95.1%) true positives. However, there are 176 (3.9%) instances misclassified as “fallen” and 46 (1%) misclassified as “orientation unk”. For the “fallen” class, the model correctly identifies one-hundred-fifty-eight (63.2%) instances. Still, there are eighty-nine (35.6%) instances misclassified as “standing” and three (1.2%) misclassified as “orientation unk”. The accuracy for the “fallen” class is reasonable, but there is room for improvement, especially in reducing misclassifications as “standing”. For the “orientation unk” class, the model correctly predicts thirty-seven (32.2%) instances, but there are seventy-one (61.7%) instances misclassified as “standing” and seven (6.1%) misclassified as “fallen”. The accuracy for the “orientation unk” class is lower compared to the other classes, suggesting a need for further improvement in classification for instances with unknown orientation.

In summary, the model performs well in several areas, but notable challenges remain. While the model excels at classifying the “metal” and “fiber” material attributes, it struggles with the “material unk” class, indicating the need for more data or improved feature extraction. For size classification, the model performs well with “short” and “long” gas cylinders but faces challenges with the “size unk” category, likely due to ambiguous features. Orientation classification is strong for “standing” gas cylinders but requires improvement mainly for the “orientation unk” class. These limitations highlight areas for future work, particularly in increasing data diversity and exploring new models’ capabilities in handling ambiguous and rare cases.

### 5.3. Future Research

For future research involving CylinDeRS, there are several key directions that researchers could investigate based on the limitations of the current version of the dataset. First, future efforts could focus on augmenting rare scenarios, including creating synthetic data, to mitigate data imbalance across attribute categories while also ensuring that models can handle edge cases, such as cylinders in hazardous conditions or extreme environments. Addressing the limitations in terms of model performance for highly occluded or reflective instances is also a critical avenue for improvement. Techniques like multi-view imaging, 3D reconstruction, or leveraging transformer-based architectures that can capture spatial and contextual relationships could be explored. Furthermore, pre-processing steps such as glare removal, image normalization, and advanced feature extraction could also help to reduce noise and improve performance. Additionally, models that incorporate contextual information—such as surrounding objects or background—are likely to yield improved performance, particularly in scenarios involving partially occluded cylinders or those in unusual orientations. Lastly, expanding the dataset with additional attributes (e.g., cylinder usage, pressure ratings, or markings and labels) could enable more specialized applications, such as safety monitoring or regulatory compliance, further enhancing the dataset’s utility.

## 6. Conclusions

This work introduced CylinDeRS, a domain-specific computer vision dataset specifically curated for two tasks: (a) object detection for detecting gas cylinder instances and (b) gas cylinder attribute classification for material, size, and orientation attributes. The proposed dataset consists of 7060 images, captured under varying settings and closely representing real-life situations, featuring a total of 25,269 annotated gas cylinder object instances. Furthermore, a systematic methodology for the creation of domain-specific datasets is proposed, covering the entire process from data collection and annotation to structure definition. A series of experiments were conducted using deep-learning-based frameworks from various categories to verify the practical application of CylinDeRS and provide insights into the strengths and limitations in this domain. While the baseline results are encouraging, achieving a maximum mAP of 91% for gas cylinder object detection and a maximum accuracy of 71.6% for attribute classification, the complex characteristics of gas cylinder instances pose notable challenges that need to be addressed. Future work includes expanding the dataset with challenging classes by adding new attribute categories and exploring performance improvements for deep learning models.

## Figures and Tables

**Figure 1 sensors-25-01016-f001:**
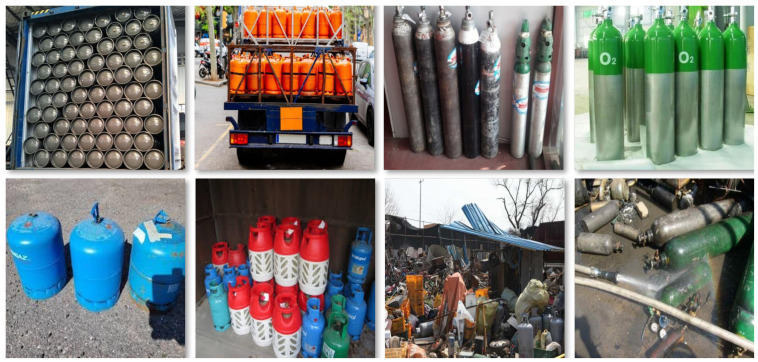
Sample images from the proposed CylinDeRS dataset, acquired under CC BY 4.0 license: (top left to bottom right) images 1–2 [34], images 3–4 [35], images 5–6 [36], image 7 [37], and image 8 [38].

**Figure 2 sensors-25-01016-f002:**
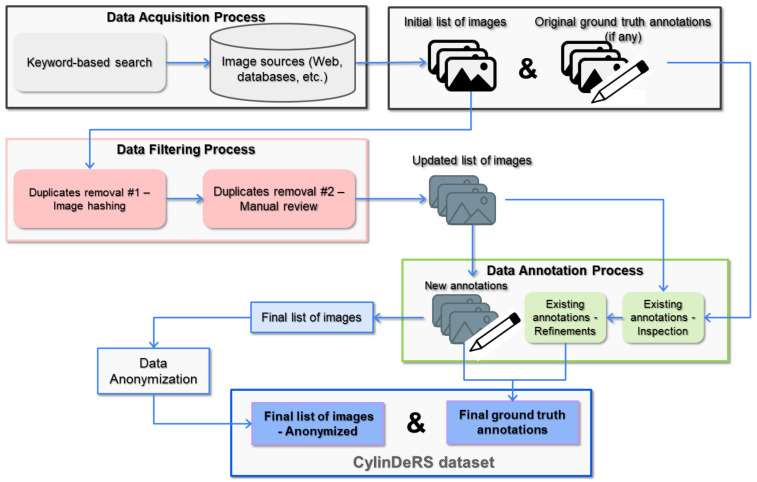
The step-by-step presentation of the proposed dataset creation methodology. Four main phases are illustrated: data acquisition, filtering, annotation, and anonymization, all contributing to the creation of the final list of images and annotations.

**Figure 3 sensors-25-01016-f003:**
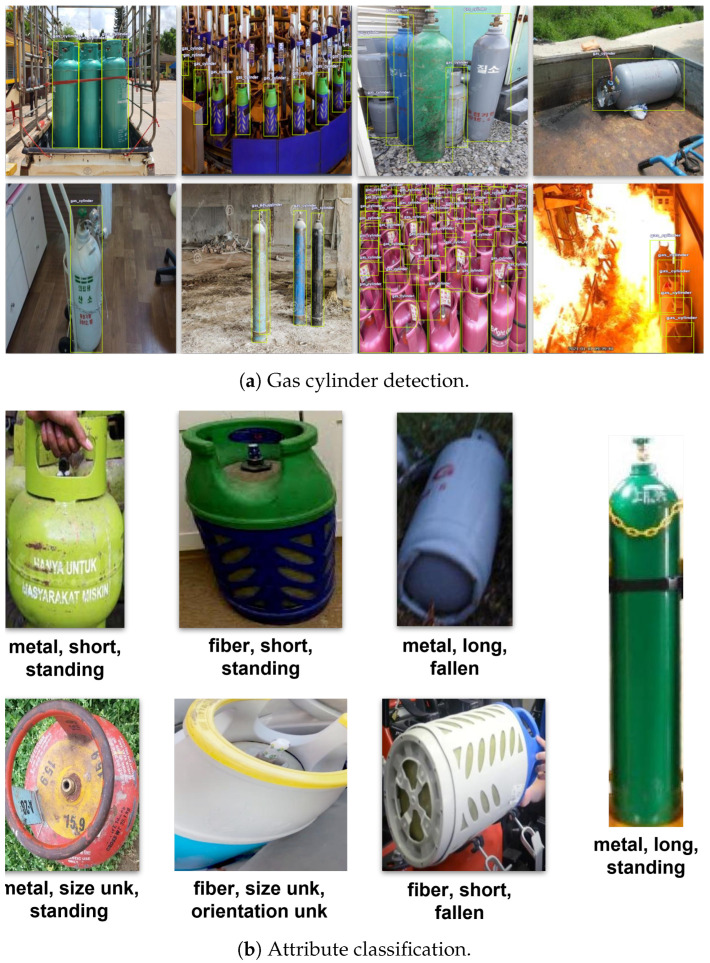
CylinDeRS dataset samples regarding (**a**) gas cylinder detection, where images are depicted with the corresponding ground truth bounding boxes, and (**b**) gas cylinder attribute classification, where gas cylinder instances are listed with their material (fiber, metal, or unknown), size (short, long, or unknown), and orientation (standing, fallen, or unknown) attribute labels.

**Figure 4 sensors-25-01016-f004:**
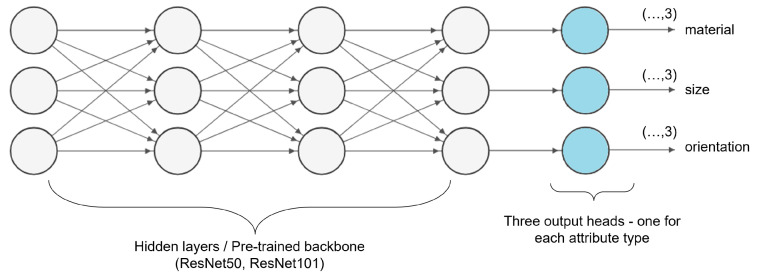
Multi-head multi-class classification in CylinDeRS, wherein each output head corresponds to a distinct attribute type, extracting a label from the three available categories for each type.

**Figure 5 sensors-25-01016-f005:**
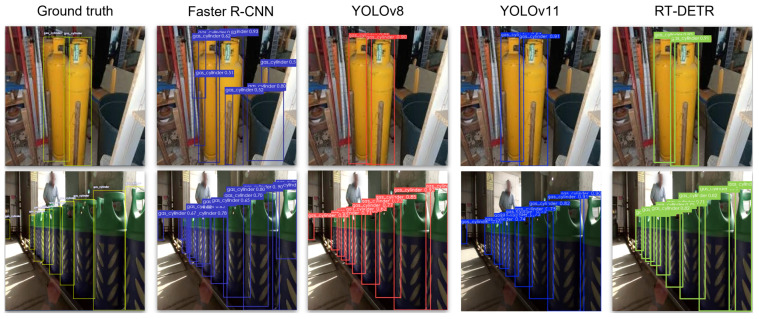
Visual representation of indicative results for the gas cylinder detection task using the CylinDeRS dataset. The left column displays the input images fed into the models with the ground truth annotation of bounding boxes colorized with yellow, while the subsequent columns showcase the corresponding detection outcomes for each of the fine-tuned models.

**Figure 6 sensors-25-01016-f006:**
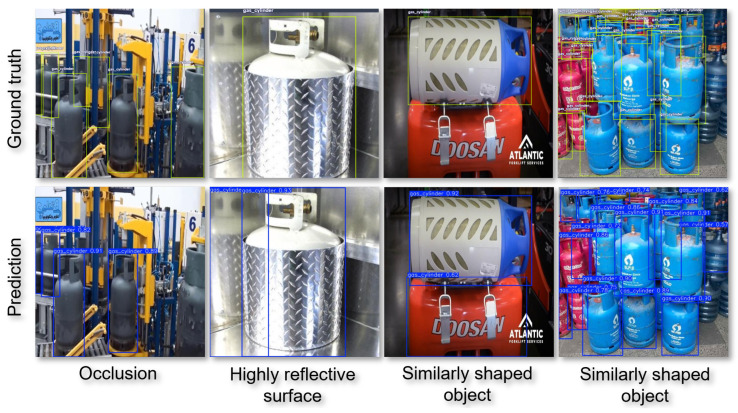
Cylinder detection false positive and negative samples in CylinDeRS dataset. The ground truth labels are depicted in the first row, while the samples in the second row indicate visual instances where the models have yielded erroneous predictions. In the last row, the potential cause of the erroneous prediction is provided.

**Figure 7 sensors-25-01016-f007:**
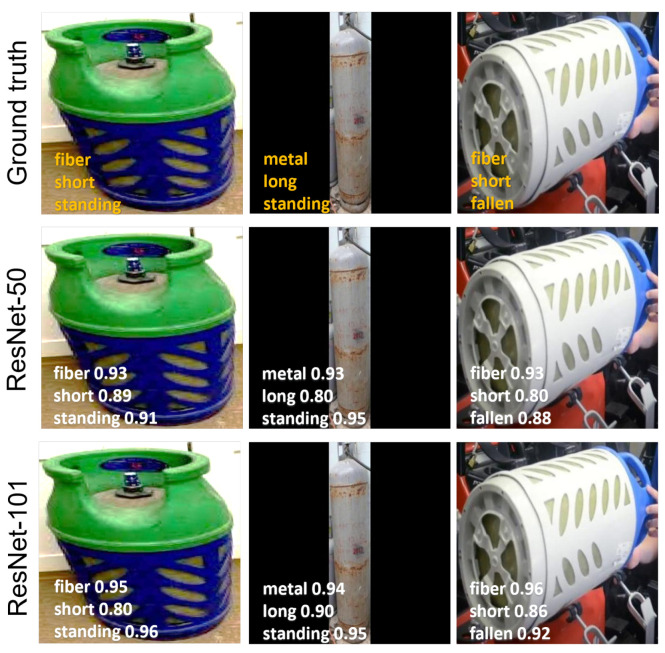
Attribute classification results on CylinDeRS dataset: representative true positive results. The top row displays images with their corresponding ground truth attribute labels. The middle and bottom rows show the predictions for the ResNet-50 and ResNet-101 models, respectively.

**Figure 8 sensors-25-01016-f008:**
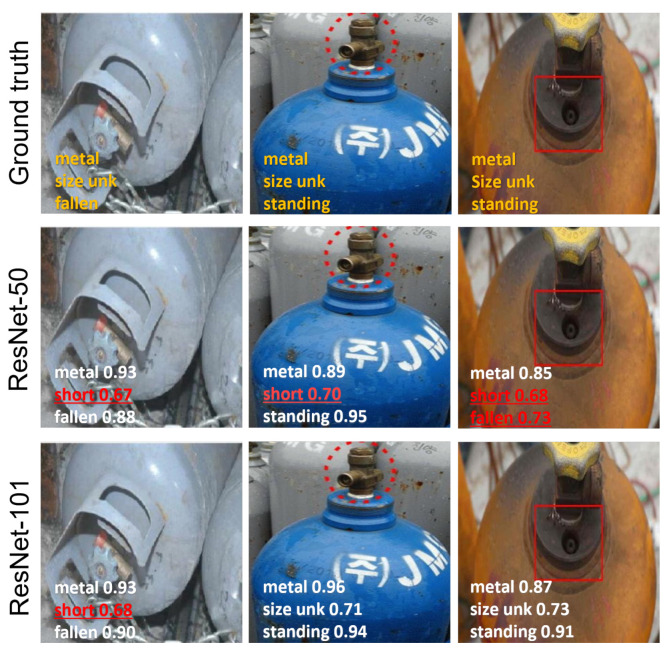
Attribute classification results on CylinDeRS dataset: representative erroneous predictions. The ground truth attribute labels are depicted in the first row. The middle and bottom rows show the predictions for the ResNet-50 and ResNet-101 models, respectively. Erroneous predictions are highlighted in red.

**Figure 9 sensors-25-01016-f009:**
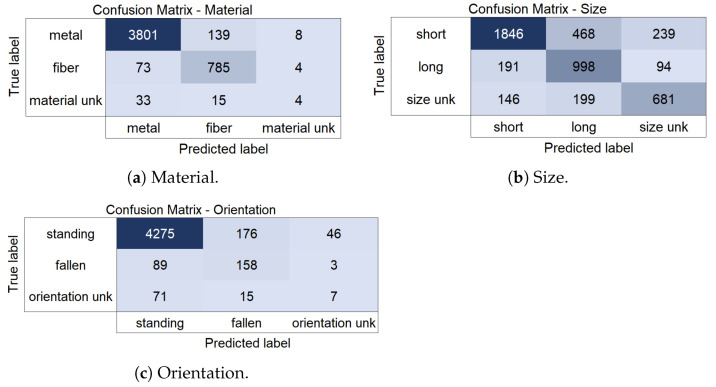
Confusion matrices for the gas-cylinder-related attributes: (**a**) material, (**b**) size, and (**c**) orientation attributes.

**Table 1 sensors-25-01016-t001:** CylinDeRS dataset statistics: number of images, number of gas cylinder instances, and average number of gas cylinders per image for the training, validation, test, and overall sets.

	Train	Val	Test	Total
No. of Images	4915	1434	711	7060
No. of Instances	18,137	4862	2270	25,269
Avg. No. of Instances per Image	3.7	3.3	3.2	3.6

**Table 2 sensors-25-01016-t002:** CylinDeRS dataset attribute category statistics: number of instances per attribute category for the training, validation, test, and overall sets.

Attribute	Category	Train	Val	Test	Total
Material	metal	14,769 (72%)	3948 (19%)	1797 (9%)	20,514 (100%)
	fiber	3201 (71%)	862 (19%)	455 (10%)	4518 (100%)
	material unk	167 (70%)	52 (22%)	18 (8%)	237 (100%)
Size	short	9705 (72%)	2553 (19%)	1192 (9%)	13,450 (100%)
	long	4598 (70%)	1283 (20%)	647 (10%)	6528 (100%)
	size unk	3834 (72%)	1026 (19%)	431 (9%)	5291 (100%)
Orientation	standing	16,688 (72%)	4497 (19%)	2126 (9%)	23,311 (100%)
	fallen	1028 (74%)	250 (18%)	101 (8%)	1379 (100%)
	orientation unk	421 (73%)	115 (20%)	43 (7%)	579 (100%)

## Data Availability

The CylinDeRS dataset is publicly available from the Roboflow repository (https://universe.roboflow.com/klearchos-stavrothanasopoulos-konstantinos-gkountakos-6jwgj/cylinders-iaq6n (accessed on 17 January 2025)). The trained models will be released upon publication of this work (https://m4d.iti.gr/results/ (accessed on 17 January 2025)).

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
