# Peer review of "CylinDeRS: A Benchmark Visual Dataset for Robust Gas Cylinder Detection and Attribute Classification in Real-World Scenes"

_sensors, 2025, doi:10.3390/s25041016_

Round 1
Reviewer 1 Report
Comments and Suggestions for Authors
Generally, the dataset paper is valuable for cylin management applications. But the manuscript is so long. I suggest that the authors simplify it and keep the main content. If there are still some useful material, they can be appendix or supplementary. On 2024Dec19th, the reviewer can not access the dataset weblink with showing "Sorry, the cylinders-iaq6 dataset does not exist, has been deleted, or is not shared with you."
minor comments:
1 In your abstract, cylin detection accuracy should also be given.
2 previous dataset introduction could be shorter
3 the 4.1, title and the content are not consistent. Faster-RCNN, YOLO, RT-DETR should not introduced with so many words
4 Section 4 and Section 5 should be combined as one section.
5 "cylin" or "gas cylin"? fix one name.
6 text in Figure 9 is small which is hard to read. redraw it
Author Response
Thank you for your constructive feedback. We agree with your suggestions and we have revised the manuscript to address your concerns. Please find below a detailed response to each of your comments.
Comment 1: "Generally, the dataset paper is valuable for cylin management applications. But the manuscript is so long. I suggest that the authors simplify it and keep the main content. If there are still some useful material, they can be appendix or supplementary."
Response 1: Thank you for pointing this out. In response, we tried to streamline the manuscript by emphasizing the core content in the main sections and addressing the reviewer’s suggestions for the Related Work section and Section 4.1 (Reviewer 1, Comments 4 and 5); you may find the relevant responses below. The manuscript has been reduced to 23 pages, down from the original 25.
Comment 2: "On 2024Dec19th, the reviewer can not access the dataset weblink with showing 'Sorry, the cylinders-iaq6 dataset does not exist, has been deleted, or is not shared with you.' "
Response 2: Thank you for pointing this out. This was mainly caused due to our initial assumption that the manuscript was still handled by the editor and not yet shared for review. The repository information was being finalized to enhance the dataset’s accessibility and usability in anticipation of the manuscript’s review. The link has been updated (Page 2, Line 62), and the CylinDeRS dataset is now publicly accessible at: https://universe.roboflow.com/klearchos-stavrothanasopoulos-konstantinos-gkountakos-6jwgj/cylinders-iaq6n.
Comment 3: "In your abstract, cylin detection accuracy should also be given."
Response 3: Thank you for this suggestion. For the gas cylinder detection task, we have chosen to report mean Average Precision (mAP), as it is a widely adopted and fundamental metric for evaluating object detection accuracy. mAP incorporates key sub-metrics such as Intersection over Union (IoU), Recall, and Precision, providing a comprehensive assessment of model performance. Reporting accuracy alone could be misleading in our case, as the dataset contains a single class, 'gas cylinder'. Accuracy would also fail to account for critical aspects, such as the precise location of detected objects within the image, which are captured through IoU.
Comment 4: “Previous dataset introduction could be shorter.”
Response 4: We agree with the reviewer and have condensed the section discussing previous datasets by summarizing the less relevant details. This revision can be found in Section 3 (Page 4, Lines 127-148). The Related Work section has been reduced from 76 to 46 lines.
Comment 5: “The 4.1, title and the content are not consistent. Faster-RCNN, YOLO, RT-DETR should not be introduced with so many words.”
Response 5: Thank you for your valuable comment. We have streamlined the descriptions of Faster R-CNN, YOLO, and RT-DETR to highlight the key aspects relevant to the study, ensuring better alignment with the section title. The revised Section 4.1 is now on Page 11, Lines 402-416.
Comment 6: “‘cylin’ or ‘gas cylin’? fix one name.”
Response 6: We appreciate this observation. To maintain consistency, we have replaced all occurrences of "cylinder" with "gas cylinder" throughout the manuscript.
Comment 7: “Text in Figure 9 is small, which is hard to read. Redraw it.”
Response 7: Thank you for pointing this out. We have redrawn Figure 9 with larger text for improved clarity. The updated figure has been included in the revised manuscript on Page 19.
Reviewer 2 Report
Comments and Suggestions for Authors
The introduction of the CylinDeRS dataset addresses a critical need in the field of computer vision, focusing on gas cylinder detection and classification. This is especially relevant given the increasing concerns around safety and illegal trade in hazardous materials.
The dataset’s size (7,060 RGB images with over 25,250 annotated instances) and its detailed annotations provide a robust foundation for training deep learning models, making it a significant resource for both academia and industry.
The potential applications of this dataset in enhancing safety protocols in industrial settings and combating environmental crimes related to hazardous substances highlight its broader societal impact.
The extensive evaluation using state-of-the-art models provides meaningful insights into the dataset's effectiveness and sets a benchmark for future research, helping to guide advancements in gas cylinder detection technologies.
The systematic methodology outlined for dataset creation—from data acquisition to annotation—serves as a valuable guideline for researchers looking to develop similar domain-specific datasets, promoting best practices in the field.
#Limitations:
It would be valuable for the authors to address any challenges encountered during dataset creation or limitations of the dataset that future researchers should consider when applying it.
The paper could benefit from discussing future research directions or potential improvements to the dataset, such as incorporating more diverse scenarios or further refining the attribute classifications.
Author Response
Thank you for your constructive feedback. We agree with your suggestions and we have revised the manuscript to address your concerns. Please find below a detailed response to each of your comments.
Comment 1: It would be valuable for the authors to address any challenges encountered during dataset creation or limitations of the dataset that future researchers should consider when applying it.
Response 1: We thank the reviewer for this insightful suggestion. A new section addressing the challenges encountered during dataset creation and the limitations of the dataset has been incorporated into the manuscript. The revisions can be found in Section 3.1.5 (Pages 7-8, Lines 281-299.
Limitations are also discussed along with the future research directions in the newly added Section 5.3, Pages 20-21, Lines 702-718.
Comment 2: The paper could benefit from discussing future research directions or potential improvements to the dataset, such as incorporating more diverse scenarios or further refining the attribute classifications.
Response 2: We appreciate the reviewer’s suggestion. To address this, we have added Section 5.3 titled “Future Research” (Pages 20-21, Lines 702-718), which outlines future research directions and potential improvements for the dataset.
Reviewer 3 Report
Comments and Suggestions for Authors
The article is well-written, the proposed database acquisition methodology is presented in exhaustive detail, and a suitable validation was performed by comparing three approaches using appropriate metrics. To further improve the work I suggest addressing the following minor issues:
- Line 64. The database is probably not available yet because the article is being revised, but please check that the link is actually working because the project is not found at the moment.
- Line 67. Which software was used to tag the images? For a more complete description, I would suggest specifying it.
- Line 268. Typo: there is a double period.
- Line 423. It would be interesting to test whether better performance is achieved with the latest version of YOLO, YOLOv11, released in 2024. This could also help understand whether action needs to be taken on the database or the model to further improve the current results.
- Line 747. What do the authors recognize as areas for improvement in the work? I would recommend expanding the section on future developments to properly clarify the directions in which they are supposed to proceed.
Author Response
Thank you for your constructive feedback. We have reviewed your useful comments and we have revised the manuscript to address them. Please find below a detailed response to each of your comments.
Comment 1: Line 64. The database is probably not available yet because the article is being revised, but please check that the link is actually working because the project is not found at the moment.
Response 1: Thank you for pointing this out. This was mainly caused due to our initial assumption that the manuscript was still handled by the editor and not yet shared for review. The repository information was being finalized to enhance the dataset’s accessibility and usability in anticipation of the manuscript’s review. The link has been updated (Page 2, Line 62), and the CylinDeRS dataset is now publicly accessible at: https://universe.roboflow.com/klearchos-stavrothanasopoulos-konstantinos-gkountakos-6jwgj/cylinders-iaq6n.
Comment 2: Line 67. Which software was used to tag the images? For a more complete description, I would suggest specifying it.
Response 2: We thank the reviewer for this suggestion. We have added the name of the online annotation tool, Roboflow Annotate, to the manuscript. It can be found on Page 3, Line 82.
Comment 3: Line 268. Typo: there is a double period.
Response 3: Thank you for noticing this typo. It has been corrected in the manuscript. The updated text can be found on Page 7, Line 242.
Comment 4: Line 423. It would be interesting to test whether better performance is achieved with the latest version of YOLO, YOLOv11, released in 2024. This could also help understand whether action needs to be taken on the database or the model to further improve the current results.
Response 4: We appreciate your suggestion. We have conducted additional experiments using YOLOv11. The results of these experiments are now included in the revised manuscript, along with a discussion of the findings. These updates can be found in Section 4, Page 11 at Lines 400-417, in Section 5, Pages 14 and 15 at Lines 550, 557-564, 580-581, Page 15 at Table 3, Figure 5, and Page 16 at Figure 6.
Comment 5: Line 747. What do the authors recognize as areas for improvement in the work? I would recommend expanding the section on future developments to properly clarify the directions in which they are supposed to proceed.
Response 5: Thank you for your useful comment, actually this part was missing from our research as it has also been observed by Reviewer 2 (Comment 2) and Reviewer 4 (Comment 4). A new Section 5.3 has been added to the manuscript on pages 19 and 20, Lines 702–718, which outlines areas for improvement and clarifies the intended future directions for the CylinDeRS dataset.
Reviewer 4 Report
Comments and Suggestions for Authors
This paper introduces CylinDeRS, a domain-specific dataset that advances gas cylinder detection and attribute classification in real-world scenes. The paper details a systematic methodology for dataset creation, including data collection, filtering, annotation, and ethical considerations. Two primary tasks are addressed: object detection and attribute classification. The language of the paper is concise, and its structure is clear. However, there are still some areas for improvement:
1. The introduction of the CylinDeRS dataset is commendable. However, the paper could further elaborate on how this dataset compares to existing datasets in terms of scale, diversity, and complexity.
2. The methodology section could include clearer visual aids (e.g., flowcharts or diagrams) to better illustrate the data collection and filtering process for readers.
3. The authors note the imbalance in attributes like "fallen" and "material unknown." It would strengthen the paper if potential solutions (e.g., data augmentation) were proposed or tested to mitigate this imbalance.
4. The paper identifies limitations in model performance for highly occluded or reflective instances. However, it does not discuss how future models or preprocessing steps might address these challenges effectively.
5. The models trained on CylinDeRS are evaluated only on the test set of the same dataset. Testing on external datasets or real-life deployments would provide insights into the generalizability of these models.
6. The paper does not discuss the computational costs or training times for the selected models.
Author Response
Thank you for your thoughtful and detailed feedback on our paper. We greatly appreciate your recognition of our work and your valuable suggestions for improvement. Please find below a detailed response to each of your comments.
Comment 1: The introduction of the CylinDeRS dataset is commendable. However, the paper could further elaborate on how this dataset compares to existing datasets in terms of scale, diversity, and complexity.
Response 1: Thank you for pointing this out. We have revised the Introduction section to include a discussion on the scale, diversity, and complexity of the CylinDeRS dataset. These updates can be found on Pages 2-3, Lines 64-79.
Comment 2: The methodology section could include clearer visual aids (e.g., flowcharts or diagrams) to better illustrate the data collection and filtering process for readers.
Response 2: Thank you for this valuable suggestion. In response, we have updated the manuscript to include a clearer visual aid in the form of an updated figure that illustrates the data collection and filtering process more effectively. This addition can be found in Section 3 (Page 5, Figure 2), and we hope it enhances the readers' understanding of our methodology.
Comment 3: The authors note the imbalance in attributes like "fallen" and "material unknown." It would strengthen the paper if potential solutions (e.g., data augmentation) were proposed or tested to mitigate this imbalance.
Response 3: Thank you for highlighting this important issue. We have updated the discussion in Section 4.4 (Page 13, Lines 491-495) to elaborate on the augmentation techniques applied to the dataset, emphasizing their role in addressing data imbalance for underrepresented classes such as "fallen" and "material unknown”.
Comment 4: The paper identifies limitations in model performance for highly occluded or reflective instances. However, it does not discuss how future models or preprocessing steps might address these challenges effectively.
Response 4: We sincerely thank the reviewer for the valuable and constructive feedback which has helped us strengthen this aspect of the manuscript. We recognize the importance of this aspect, which was indeed missing from our initial submission, as also highlighted by Reviewer 2 (Comment 2) and Reviewer 3 (Comment 5). In response, we have expanded our discussion to address occluded and reflective instances, incorporating these considerations in the newly added Section 5.3, “Future Research”. These updates can be found on Pages 19 and 20, Lines 702–718.
Comment 5: The models trained on CylinDeRS are evaluated only on the test set of the same dataset. Testing on external datasets or real-life deployments would provide insights into the generalizability of these models.
Response 5: Thank you for your thoughtful suggestion. The primary focus of this work is on the introduction and analysis of the CylinDeRS dataset. The models presented are intended to serve as indicative baselines, providing initial insights into the dataset's utility. While generalizability is an important aspect of model evaluation, we consider that this lies beyond the scope of this study.
Comment 6: The paper does not discuss the computational costs or training times for the selected models.
Response 6: Thank you for raising this important point. To address this, we have added a discussion in Section 4.5 detailing the training times for the selected models. The updates can be found in Page 13 Lines 511-519.
Round 2
Reviewer 1 Report
Comments and Suggestions for Authors
I recommend an acceptance to publish.